# Investigation of Rare Non-Coding Variants in Familial Multiple Myeloma

**DOI:** 10.3390/cells12010096

**Published:** 2022-12-26

**Authors:** Yasmeen Niazi, Nagarajan Paramasivam, Joanna Blocka, Abhishek Kumar, Stefanie Huhn, Matthias Schlesner, Niels Weinhold, Rolf Sijmons, Mirjam De Jong, Brian Durie, Hartmut Goldschmidt, Kari Hemminki, Asta Försti

**Affiliations:** 1Hopp Children’s Cancer Center (KiTZ), 69120 Heidelberg, Germany; 2Division of Pediatric Neurooncology, German Cancer Research Center (DKFZ), German Cancer Consortium (DKTK), 69120 Heidelberg, Germany; 3Computational Oncology, Molecular Precision Oncology Program, National Center for Tumor Diseases (NCT), 69120 Heidelberg, Germany; 4Department of Internal Medicine V, University of Heidelberg, 69120 Heidelberg, Germany; 5Department of Medical Oncology, Jerome Lipper Multiple Myeloma Center, Dana-Farber Cancer Institute, Boston, MA 02115, USA; 6Harvard Medical School, Boston, MA 02115, USA; 7Institute of Bioinformatics, International Technology Park, Bangalore 560066, India; 8Manipal Academy of Higher Education (MAHE), Manipal 576104, India; 9National Center for Tumor Diseases Heidelberg (NCT), 69120 Heidelberg, Germany; 10Bioinformatics and Omics Data Analytics, German Cancer Research Center (DKFZ), 69120 Heidelberg, Germany; 11University Medical Center Groningen, University of Groningen, 9712 Groningen, The Netherlands; 12Cedars Sinai Cancer Center, Los Angeles, CA 90048, USA; 13Division of Cancer Epidemiology, German Cancer Research Center (DKFZ), 69120 Heidelberg, Germany; 14Faculty of Medicine and Biomedical Center in Pilsen, Charles University in Prague, 323 00 Pilsen, Czech Republic

**Keywords:** non-coding genome, familial multiple myeloma, MAPK pathway, whole-genome sequencing

## Abstract

Multiple myeloma (MM) is a plasma cell malignancy whereby a single clone of plasma cells over-propagates in the bone marrow, resulting in the increased production of monoclonal immunoglobulin. While the complex genetic architecture of MM is well characterized, much less is known about germline variants predisposing to MM. Genome-wide sequencing approaches in MM families have started to identify rare high-penetrance coding risk alleles. In addition, genome-wide association studies have discovered several common low-penetrance risk alleles, which are mainly located in the non-coding genome. Here, we further explored the genetic basis in familial MM within the non-coding genome in whole-genome sequencing data. We prioritized and characterized 150 upstream, 5′ untranslated region (UTR) and 3′ UTR variants from 14 MM families, including 20 top-scoring variants. These variants confirmed previously implicated biological pathways in MM development. Most importantly, protein network and pathway enrichment analyses also identified 10 genes involved in mitogen-activated protein kinase (MAPK) signaling pathways, which have previously been established as important MM pathways.

## 1. Introduction

Multiple myeloma (MM) is a malignancy of plasma cells that are specialized and terminally differentiated B cells. Plasma cells synthesize and secrete antibodies to maintain humoral immunity. MM is characterized by the expanded proliferation of a single clone of plasma cells in the bone marrow, leading to the enhanced production of monoclonal immunoglobulin, also called M protein. The presence of M protein is an important diagnostic criterion for MM, along with the “CRAB” features, which is a mnemonic for calcium levels, renal failure, anemia and bone lesions, which have recently been extended [1]. In most cases, patients diagnosed with MM have one of the two precursor conditions, monoclonal gammopathy of unknown significance (MGUS) or smoldering multiple myeloma (SMM) [1].

MM is the second most common hematological malignancy, responsible for 1% of overall cancer-related deaths [2]. Although a relatively uncommon global disease, it is prevalent in countries with high socioeconomic status [3]. The genetic architecture of multiple myeloma is very complex. It consists of primary and secondary genetic events, including, but not limited to, chromosomal translocations, regional gains and deletions, hyperdiploidies, gene mutations and copy number variations (CNVs) [1]. In addition, high-risk, rare, high-penetrance germline variants have been discovered through whole-exome (WES) and whole-genome sequencing (WGS) in MM families [4,5,6]. Genome-wide association studies (GWASs) have also helped to discover several common and low-penetrance risk loci [7,8].

Inherited predisposition to MM is evident among the first-degree relatives of MM patients, who are at a 2–4 times higher risk of developing this disease when compared to the general population [9]. In our previous study, we investigated 21 MM/MGUS families to identify germline predisposition genes through WGS and WES. Several pathogenic coding variants, including missense, loss-of-function (LoF) and CNVs, were identified. These variants were in genes functionally related to previously suggested MM susceptibility, immune process, tumor-related and MM somatic driver genes [6].

To further explore the basis of MM predisposition in MM families, we focused on the non-coding region of the genome in the present study. The non-coding region makes up 98% of the total human genome. Moreover, non-coding variants are gaining importance in the understanding of inherited cancer susceptibility [10]. Non-coding variants, e.g., the 5′ untranslated region (UTR) and 3′ UTR, due to their location upstream of the transcription start and downstream of the transcription end site, respectively, can bring about changes in transcription and posttranscriptional regulation. Considering the meaningful regulatory potential of these variants, we examined and prioritized non-coding variants from the WGS data of 14 MM families from Germany and the Netherlands. Prioritization was carried out using our internally established Familial Cancer Variant Prioritization Pipeline (FCVPPv2) [11] and other non-coding variant prioritization tools, such as Combined Annotation Dependent Depletion v1.6 (CADD) and SNPnexus [12,13].

## 2. Materials and Methods

### 2.1. Multiple Myeloma Families and Whole-Genome Sequencing

Samples from the patients and their healthy family members, as well as other familial and clinical information, were obtained after informed written consent. The study was carried out according to the rules of the Declaration of Helsinki, after the approval of the ethics committee of the Medical Faculty of the University of Heidelberg. All the patients, from the University Medical Center Groningen (UMCG), Netherlands, were briefed and signed consent was obtained for WGS to identify the cause of cancer predisposition in their families. These patients were enrolled as part of the Groningen-Heidelberg-Stettin EU TRANSCAN familial cancer whole-genome sequencing project because of their family history of cancer. They were referred to UMCG clinically for diagnostics and counseling because of their cancer family history. Clinical requirements for their testing and WGS did not indicate any further need for the approval of the ethics review board of the UMCG.

In total, 14 families with 31 cases and 16 unaffected individuals (controls or possible carriers) participated in this study (Appendix A). Among these, 12 families were recruited from Heidelberg, Germany, and two from the Netherlands. At least two cases were enrolled from each family. These individuals were diagnosed either with MM and its precursors, MGUS and SMM, or with AL amyloidosis. Participating unaffected family members recruited in Heidelberg were analyzed for the following parameters: blood count, creatinine, glomerular filtration rate, calcium, immunoglobulin levels, free light chains and their ratios, protein electrophoresis and immunofixation in serum and urine to exclude undetected MM or its precursor stages [14]. Only individuals with negative immunofixation in serum and urine were considered unaffected.

Sequencing of all the samples was carried out at the core facility of DKFZ in Heidelberg. DNA was extracted from peripheral blood using the QIAamp^®^ DNA Mini Kit (40724 Hilden, Germany). Paired-end sequencing with a 150 bp read length was performed on the Illumina X10 platform (10785 Berlin, Germnay), followed by sequence mapping to the reference human genome (build GRC37, assembly hs37d5) using BWA mem (version 0.7.15, with parameters: –T 0) [15] and the removal of duplicates via Sambamba (version 0.6.5, with parameters: t 1 -l 0 --hash-table- size = 2000000 --overflow-list-size = 1000000 --io-buffer-size = 64) [16].

Variant calling for single-nucleotide variants (SNVs) and indels was carried out using Platypus (version 0.8.1) [17]. The variants were annotated with Gencode (v19) gene definitions in a multistep process using the following tools: ANNOVAR [18], 1000 Genomes phase III [19], dbSNP [20], dbNSFP v2.9 [21] and ExAC [22] at a read depth of >10. A minor allele frequency threshold of 0.001 was used for gnomAD exome and genome data [23] and a variant frequency of 2% from the local set to remove common variants and technical artifacts, respectively. A pairwise comparison of the variants in the cohort was performed to confirm family relatedness and exclude sample mix-ups.

### 2.2. Prioritization through FCVPPv2

We used our in-house variant filtering pipeline, the Familial Cancer Variant Prioritization Pipeline (FCVPP) version 2, developed by Kumar et al., for the pedigree-based prioritization of the variants [11]. Pedigree segregation meant that variants were selected if they were present in all the cases of a family and absent from all the healthy family members. The possible carriers could show either the presence or absence of the variant of interest. Family members were considered as cases if they were diagnosed with MM, MGUS or AL amyloidosis. Those detected with plasma cell dyscrasias, solitary plasmacytomas and aberrant plasma cell clones were termed “possible carriers”. Healthy family members without these two parameters (MM, MGUS or AL amyloidosis diagnosis and plasma cell anomalies) were considered non-carriers. The exceptions to the above rule were the healthy family members who were more than 10 years younger than the earliest age of diagnosis in the family; these were treated as “possible carriers”. Using the CADD tool v1.3, a filter of ≥15 was applied after pedigree segregation to obtain the top 1.5% deleterious variants in the human genome. In addition, another web-based annotation tool, SNPnexus [13], was used to check for different non-coding scores, such as EIGEN [24], Funseq2 [25], FATHMM [26], ReMM [27] and Deep-SEA [28].

After these filtering steps, non-coding variants were selected for further evaluation; these included 5′ UTR, 3′ UTR, upstream variants and variants that were labeled upstream and downstream. The variants were visually inspected, using the Integrative Genomics Viewer (IGV; version 2.4.10) [29], within WGS data for cases and controls, as an added measure to minimize the possibility of false-positive results and to enhance the confidence of variant calls.

### 2.3. Conservation

The selected non-coding variants were then prioritized based on their conserved locations using three different evolutionary conservation scores; these included Genomic Evolutionary Rate Profiling (GERP) score > 2 [30], vertebrate PhastCons ≥ 0.3 [31] and vertebrate Phylogenetic P-value (PhyloP) ≥ 3.0 [32]. Variants were additionally assigned a score of 1–3 depending upon how many out of the three conservation scores were positive.

### 2.4. Analysis of Upstream and 5′ UTR Variants

The 5′ UTR and upstream variants were investigated according to the following steps. At first, the variants were intersected with the human promotor database downloaded from FANTOM 5 [33] using bedtools. CADD v1.6′s web-based interface gave information about the percentage of GC content, presence of CpG islands, transcription factor binding sites (TFBSs) and chromatin states in 127 cell lines and histone marks in 14 cell lines and tissues for the loci that our variants were present in.

### 2.5. TFs/TF Binding Sites

Prioritized upstream and 5′UTR variants were further assessed based on their location at TFBSs. Publicly available TF ChIP-seq data were obtained from ENCODE for the GM12878 cell line [34]. These data were compared with previously published TF enrichment data for MM [7]. To investigate the effect of a variant on TF binding, short FASTA mutated and wild-type sequences having variant points with 10 bp upstream and 10 bp downstream were uploaded on JASPAR for the above-mentioned best-performing variants [35].

### 2.6. Graphic Visualization

To obtain a visual representation of 5′ UTR and upstream variants along with the different regulatory elements, variant maps were created using the UCSC genome browser [36].

### 2.7. Analysis of 3′ UTR Variants

The 3′ UTR variants were further investigated for being located at putative miRNA target sites. For this purpose, the entire human miRNA target atlas was downloaded from TargetScan (Release 7.0) [37] and matched against the filtered 3′ UTR variants using bedtools’ intersect function to obtain miRNA matches along with a context++ percentile score. A context++ score percentile of 90 or above was considered to be a significant score. Using CADD v1.6. [12], ChromHmm chromatin states (from 127 cell lines from the NIH roadmap epigenomics mapping consortium) [38], the Segway chromatin pattern [39] and the mirSVR score were extracted. Variants were marked positively if they had a mirSVR score of less than −0.1, as sites with mirSVR scores lower than −0.1 are generally considered good miRNA target sites with a high probability of downregulation of gene expression [40].

### 2.8. Biological Function and Pathway Enrichment Analysis

All the respective genes from the pipeline surviving variants were used for protein interaction network analysis using STRING v10 [41] and for pathway enrichment analysis using Reactome [42]. Biological function information for both sets of variant genes was collected through UniProtKB/Swiss-Prot [43].

A sequential flow chart of all the above prioritization tools with the filtered number of variants at each step is shown in Figure 1.

## 3. Results

WGS on 14 MM families identified 928,170 rare variants (MAF < 0.1%); these included variants annotated by ANNOVAR as exonic, intronic, intergenic, splicing, upstream, downstream, upstream; downstream, 3′ UTR, 5′ UTR and 3′ UTR; 5′ UTR (Figure 1). Among these annotations, the focus of this present work was on the 3′ UTR, 5′ UTR and upstream variants, which amounted to 20,445. After pedigree segregation in the next step, this number was reduced to 2682. Further pruning was performed when the CADD score of ≥ 15 was applied, resulting in 150 variants. As most of the non-coding scores extracted using SNPnexus were high after filtering for CADD ≥ 15 (data not shown), these were not used for the prioritization of the variants. Out of these pipeline-surviving variants, 51 were 5′ UTR, 53 were upstream or upstream; downstream and 46 were 3′ UTR variants.

Through the in silico functional analysis of the 104 5′ UTR and upstream variants with CADD v1.6., a conservation score, the presence in the promotor region of the respective gene and within a CpG island, as well as the chromatin state, histone marks and TFBSs on the location of each variant were compiled, as shown in Appendix A, and all TFs binding to the variant positions according to the ENCODE data are shown in Appendix A. The variants with positive scores of the selected annotations in CADD v1.6. were shortlisted as the 14 top variants (Table 1). Genes identified through the top variants were *SP5* (transcription factor Sp5), *FNDC3B* (fibronectin type III domain containing 3B), *FOXJ2* (forkhead box protein J2), *NRBF2* (nuclear receptor binding factor 2), *HMGXB4* (HMG box domain containing 4), *AGFG1* (ArfGAP with FG repeats 1), *ING2* (inhibitor of growth family member 2), *MDFIC* (MyoD family inhibitor domain containing), *TBC1D4* (TBC1 domain family, member 4), *ERBB3* (v-erb-b2 erythroblastic leukemia viral oncogene homolog 3 (avian)), *PSMC6* (proteasome (prosome, macropain) 26S subunit, ATPase, 6), *CAMK1* (calcium/calmodulin-dependent protein kinase I), *PLEKHG1* (pleckstrin homology domain containing, family G (with RhoGef domain) member 1) and *DLG1* (discs, large homolog 1 (Drosophila)). All these top variants were annotated to be in the promoters of the corresponding genes, except the variant in *DLG1*, which was annotated to the promoter of *DLG1-AS1*. All were also mapped to CpG islands and they were located within binding sites for many important TFs, as shown in Table 1 and Appendix A, extracted through ENCODE [34]. A FASTA sequence search around the variant through Jaspar [35] also showed the changes in binding sites due to these variants (Appendix A). Limited consensus was observed between TFBSs from ENCODE and Jaspar; in most cases, both highlighted the same TF families. We here only show the TFBS differences caused by our variants between the wild-type and mutated sequence in the Jaspar tables. Segway classification, chromatin state and histone mark evaluation supported their locations in active transcription start sites and in promotor or enhancer regions (Table 1, Appendix A).

The in silico functional scores for the 46 3′ UTR variants are described in Appendix A, including conservation scores, miRNA binding sites, mirSVR scores and chromatin states. Appendix A presents all miRNAs binding to the positions of these variants. The variants with positive scores in all aspects of functional analysis were shortlisted as six 3′ UTR top variants. Among these top shortlisted variants, we identified genes such as *LONRF1* (LON peptidase N-terminal domain and ring Finger 1), *SGSM2* (small G protein signaling modulator 2), *SLC35A1* (solute carrier family 35 member A1), *B4GALT5* (beta 1,4-galactosyltransferase, polypeptide 5), *MARCHF8* (membrane-associated ring finger 8, E3 ubiquitin-protein ligase) and *FAM76B* (family with sequence similarity 76, member B). All six variants fulfilled the conservation criteria, all had good mirSVR scores (<−0.1), and Segway and chromatin marks confirmed the location at the gene end (Table 2). miRNA matches were found for all of the selected variants, and, except for miRNAs in *B4GALT5* and *MARCHF8*, all others had very high context++ scores (>90).

In a couple of instances, two different variants of the same gene were prioritized in unrelated families. One of these genes was *CAMK1* (calcium/calmodulin-dependent protein kinase I). Its variants were identified in two families, i.e., in family 15, a 5′ UTR variant (3_9811535_G_A), and in family 2, a 3 ‘UTR variant (3_9799045_C_G). The other was *ZNF236*, which had a 3 ’UTR variant in family 6 and a 5′ UTR variant in family 2.

The prioritized variants located in the 5′UTR and upstream or 3′UTR were grouped according to their biological functions obtained through Uniprot (Figure 2). Common pathways between both sets of genes included transcription, signal transduction, cell cycle and differentiation, bone development, metabolism and growth, chromatin organization, cell adhesion, immune system and protein transport.

Independent protein interaction network and pathway enrichment analyses for the proteins corresponding to the genes with all the 150 upstream, 5′ UTR and 3′ UTR variants were performed with STRING and Reactome pathway analyses, respectively, and they gave similar results. Figure 3 shows the STRING protein interaction network for proteins of the upstream and 5′ UTR variant genes, highlighting proteins that belong to the mitogen-activated protein kinase (MAPK) and ErbB pathways in blue and green colors, respectively. These included *PIK3R1* (phosphatidylinositol 3-kinase regulatory subunit alpha/P85-ALPHA), *DLG1* (Discs large MAGUK scaffold protein 1), *SPTB* (ppectrin beta chain, erythrocytic (Beta-I spectrin)), *APBB1IP* (amyloid beta A4 precursor protein-binding family B member 1-interacting protein), *CAMK2D* (calcium/calmodulin-dependent protein kinase type II subunit delta), *ERBB3* (Erb-B2 receptor tyrosine kinase 3), *ERBB4* (receptor tyrosine-protein kinase erbB-4), *PSMC6* (proteasome 26S subunit, ATPase 6) and *PTK2/FAK1* (protein tyrosine kinase 2/Focal adhesion kinase 1). Functional details of all these pathway variants are also included in Table 1. Among these nine genes, *ERBB3*, *PSMC6* and *DLG1* were already among the pipeline’s prioritized top variants.

Figure 4 depicts the protein interaction network for proteins corresponding to the 3′ UTR variant genes and showing no pathway enrichment. Combining both of the above-mentioned sets of proteins confirmed their involvement in the MAPK and ErbB pathways (Figure 5). These included six genes that were already identified to be involved in the MAPK pathway in the 5′ UTR analysis and only one 3′ UTR variant gene, i.e., *FGFR1* (fibroblast growth factor receptor 1), in the main core of the network.

Reactome confirmed the involvement of the MAPK and ErbB pathways (Table 3). It can be seen from the table that Reactome excludes STAT5A from the RAF/MAP kinase cascade and includes it in ERBB4 signaling but with a high false discovery rate (FDR). Additional pathway enrichment on STRING was performed by combining our gene set with the genes previously identified in our MM family study of missense and LoF variants [6] (Appendix A). This did not highlight any new pathways, and the MAPK pathway remained the only considerably enriched pathway.

Visual maps of the genetic and regulatory environment of the pipeline-prioritized top upstream and 5′ UTR variants, as well as those identified by pathway enrichment analysis from Table 1, were created using the UCSC genome browser. These maps show variation sites relative to their positions in the gene (within gene promoter or enhancer/strong or weak promoter or enhancer), the number of CpG islands, conserved sequences, histone methylation marks and TFBSs in the GM12878 cell line with the help of UCSC annotation tracks (Appendix A).

## 4. Discussion

The genetic architecture of familial MM, despite progress in global MM research, remains largely elusive. Previous GWASs on MM point towards the non-coding part of the genome influencing the gene function through regulation [7,44]. With the study of non-coding variants in the WGS data from 14 MM families, we have tried to bridge a part of the knowledge gap in MM research. We identified 150 non-coding variants including 5′ UTR, upstream and 3′ UTR variants that segregated with MM cases among families and passed the CADD ≥ 15 criterion. These variants, when grouped based on the biological function of the corresponding genes and proteins, highlighted similar pathways that have previously been implicated by risk loci in MM GWASs [7] and familial rare germline variant investigation [6]. The highlighted pathways included the immune system, chromatin remodeling, cell cycle regulation, signal transduction and autophagy (Figure 2a,b). Further protein interaction network and pathway enrichment analyses highlighted the importance of the MAPK and ErbB pathways in the germline genetics of MM.

The prioritization of the variants was based on the segregation of the variants with MM in the families, followed by the well-established pipeline to further prioritize the variants based on several in silico prediction tools. For 5′ UTR and upstream variants, several tools with different aspects of potential regulatory effects are available; however, for 3′ UTR variants, most tools concentrate on the effects due to changes in miRNA binding sites. Thus, we were not able to evaluate other factors, such as the effects of the variants on the stabilization of the termination codons or enzymatic cleavage sites. Although most of the presented candidate genes are unlikely to have a causal relationship with MM, we are convinced that our data could be a valuable contribution to forthcoming, pooled sequencing efforts. Below, we discuss potential mechanisms explaining how the identified variants and genes may predispose to MM.

The importance of signal transduction pathways in MM has already been demonstrated [45]. These pathways play an important role in the interaction between MM cells and other cellular components, such as osteoclasts, osteoblasts, dendritic cells and endothelial cells in the bone marrow microenvironment [46]. Multiple signaling cascades are activated by a vital myeloma growth factor, interleukin-6 (IL-6), including the Jak-Stat, the Ras/Raf/Mek/Erk and the PI-3′kinase/Akt pathways [47]. Ras/Raf proteins also regulate the MYC gene promoter through the Raf/MAPK/MEK pathway. The involvement of the MAPK pathway in MM pathogenicity is reaffirmed in this study through functional protein interaction network and pathway enrichment analyses of variant genes on the STRING database and Reactome. A set of 150 proteins corresponding to the upstream, 5′ and 3′ UTR variant genes highlighted RAF/MAPK and its upstream ErbB signaling pathways. The resulting network contained 10 of ~300 genes that are involved in the RAF/MAPK pathways, and 6 of 83 genes in ErbB signaling pathways. The RAF/MAPK pathway genes identified in our network were *PIK3R1, DLG1, FGFR1, SPTB, APBB1IP, CAMK2D, ERBB3, ERBB4, PSMC6* and *PTK2/FAK1* (Figure 5, Table 3). Incidentally, three of these genes, *ERBB3, PSMC6* and *DLG1*, were also among the top upstream and 5‘ UTR variant-related genes selected independently due to the all-round best performance in the different in silico functional analysis tools employed.

The ErbB family of proteins are receptor tyrosine kinases (RTKs). ErbB RTKs dimerize after the binding of ligands to their extracellular domains, leading to auto-phosphorylation, followed by the downstream signaling cascades [48]. One of the major signaling cascades of the ErbB family is the Ras/Raf/MAPK pathway [49]. A recent study evaluated the role of ERBB3/ERBB4 in signal transduction in mouse cells that expressed only ERBB3 and ERBB4. Upon enrichment analysis of regulated phosphoproteins with KEGG pathways, it was revealed that ErbB signaling, focal adhesion and MAPK signaling were among the top enriched pathways [50]. Signaling pathways downstream of RTKs have long been identified as therapeutic targets in different cancers [51]. MAPK activation through ERBB signaling controls key processes such as cellular growth, proliferation, differentiation, migration and apoptosis [52]. MAPK pathways mediate the signals that either promote or suppress the growth of malignant cells, and their critical role in the development of hematological malignancies, including multiple myeloma, has been demonstrated previously [47].

PSMC6 is a 26S proteasome subunit. Proteasome inhibition is an important part of therapy in MM patients since the efficacy of bortezomib was discovered some 20 years ago [53]. However, the development of resistance to bortezomib is common and it is now found that the downregulation of PSMC6 is one of the most common and validated reasons for conferring bortezomib resistance [54]. DLG1 is a multidomain scaffolding protein that plays a part in fundamental cellular pathways [55]. It also promotes the growth and survival of myeloma cells in bone marrow-independent niches by facilitating the interaction between CD28 and CD86 molecules on the cell surface. This allows the MM cells to be independent of the bone marrow microenvironment, resulting in extramedullary multiple myeloma (EMM) [56]. It is interesting to note that two other MAPK pathway-enriched genes, *SPTB* and *PTK2/FAK1*, also play a role in the development of an aggressive and rare form of EMM called plasma cell leukemia [57].

Fibroblast growth factor receptors, including FGFR1, are also members of the RTK family of receptors that play an important role in cell survival, differentiation, migration and proliferation. They have high homology with each other and bind to fibroblast growth factors (FGFs) [58]. Previously, we identified a CNV affecting genes *FGFBP1* (fibroblast growth factor binding protein 1) and *FGFBP2* (fibroblast growth factor binding protein 2) in a study of coding variants in MM families [6]. FGFBP1 and FGFBP2 are involved in FGF bioactivation and may affect cell proliferation and the bone microenvironment in MM. *FGFR1* is the only 3′ UTR variant gene that was highlighted in the MAPK pathway. FGFR mutations in different malignancies make them attractive targets for therapy. Recent studies show that the development of resistance to FGFR inhibitors is achieved through the activation of ERBB2 and ERBB3 [59]. An indirect adaptor-mediated interaction between FGFR1 and PIK3R1 (P85) also results in the activation-dependent regulation of extracellular-signal-regulated kinase (ERK) in MM cells [60].

CAMK2D is one of the isoforms of calcium2+/calmodulin dependent protein kinases. Calcium, as a second messenger, plays an important role in the development of B cells. Out of the four isoforms, alpha, beta, gamma and delta, the latter three are more widely expressed in the body, especially in lymphoid tissues, including bone marrow, and are involved as mediators in MAPK-dependent apoptosis pathways activated by calcium flux [61].

Other upstream and 5′ UTR variants that were among the all-round top candidates included variants in genes related to transcription regulation, such as *SP5, MDFIC, FOXJ2* and *NRBF2* [62,63,64,65]. Elevated expression of SP5 has been detected in different human cancers [63] and can downregulate many WNT target genes, resulting in a decreased transcription response [66]. *NRBF2* is involved in autophagy [65]. *PLEKHG1* is also a top variant gene related to cell signaling [67]. Another cell signaling-related gene was HMGXB4, which is also involved in the Wnt/β-catenin signaling pathway [68]. The HMGXB4-TOM1 locus has been suggested as a myeloma risk locus at 22q13 [69].

The remaining four genes in the upstream and 5′ UTR top variant gene lists were *TBC1D4, ING2, AGFG1* and *FNDC3B*, related to protein transport, mRNA transport and adipogenesis, respectively [43].

Among the top shortlisted variants in 3′ UTR, we identified genes such as LONRF1 and SGSM2, involved in protein ubiquitination and transport, respectively [70,71]. *SLC35A1* and *B4GALT5* are related to metabolism [72] and *MARCHF8* is associated with the immune system [73]. A variant in *FAM76B* is also among the best-performing variants; however, the function of this gene is unknown. A search for previously recognized MM-related miRNAs [74] in our list did not prove fruitful; however, common miRNAs were present for different gene variants in different families.

All variants were specific to each family; however, for two genes, *CAMK1* and *ZNF236*, two different variants, one in the 3′ UTR and one in the 5′ UTR, were prioritized in two unrelated families. CAMK1 plays a role in the G1 phase of the cell cycle, where it regulates the assembly of the cyclin D1/cdk4 complex [75]. Amplification of the *cyclin D1* gene has not only been associated with multi-drug resistance in MM [76] but a polymorphism in the gene is a risk factor for t(11;14)(q13;q32) MM [77]. Regarding *ZNF236*, in our previous study of rare germline variants in familial MM, a missense variant in the gene was found [6]. Because of the limited knowledge of the function of this gene, it is difficult to link the potential pathogenicity of these variants to MM. The gene is believed to play a role in transcription regulation [43]. Recently, miRNA regulation for this gene was observed in a cleft palate-associated gene study, where *ZNF236* overexpression was linked to cell proliferation [78].

In conclusion, we have identified new non-coding gene variants conferring a predisposition to MM in familial cases. Many of these variants are found in pathways and genes previously implicated in MM risk, and thus reaffirm the involvement of the ErbB and MAPK signaling pathways in MM pathogenicity. These results also highlight the importance and potential of the non-coding genome in the underlying mechanisms of different diseases.

## Figures and Tables

**Figure 1 cells-12-00096-f001:**
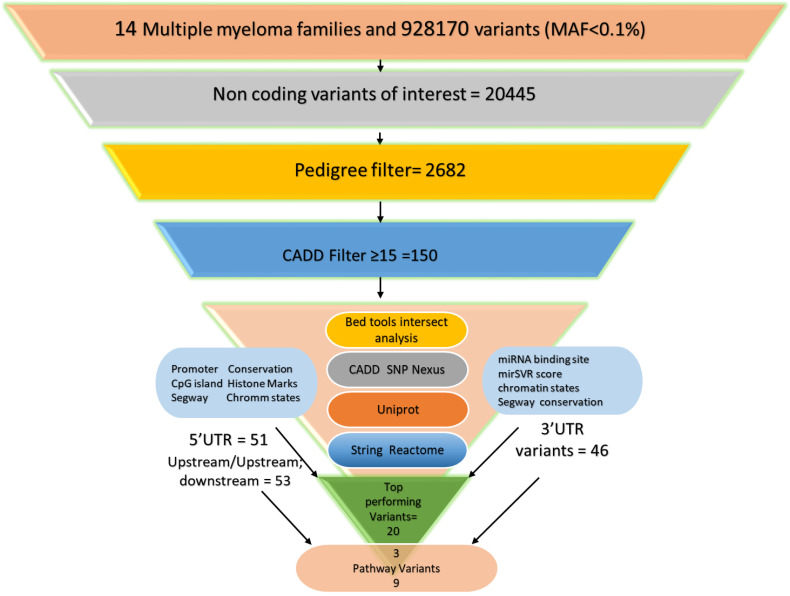
Prioritization pipeline with the number of filtered variants at each step, including the number of top variants among 5′ UTR, upstream and 3′ UTR variants, and variants identified by the protein interaction and pathway enrichment analyses.

**Figure 2 cells-12-00096-f002:**
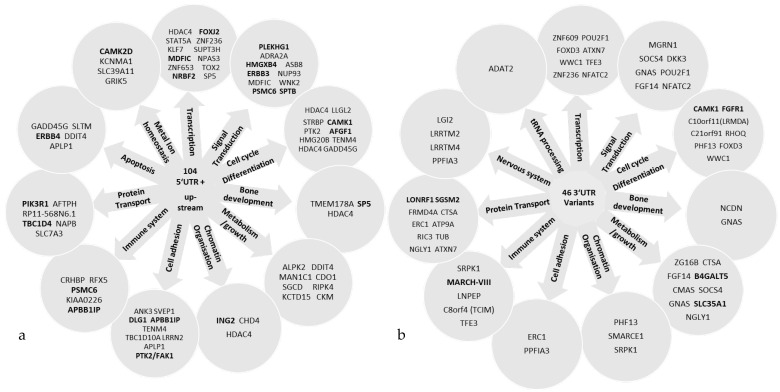
Key pathways highlighted by genes with variants in (**a**) 5′ UTR/upstream and (**b**) 3′ UTR analysis. Bold text indicates the genes with top variants.

**Figure 3 cells-12-00096-f003:**
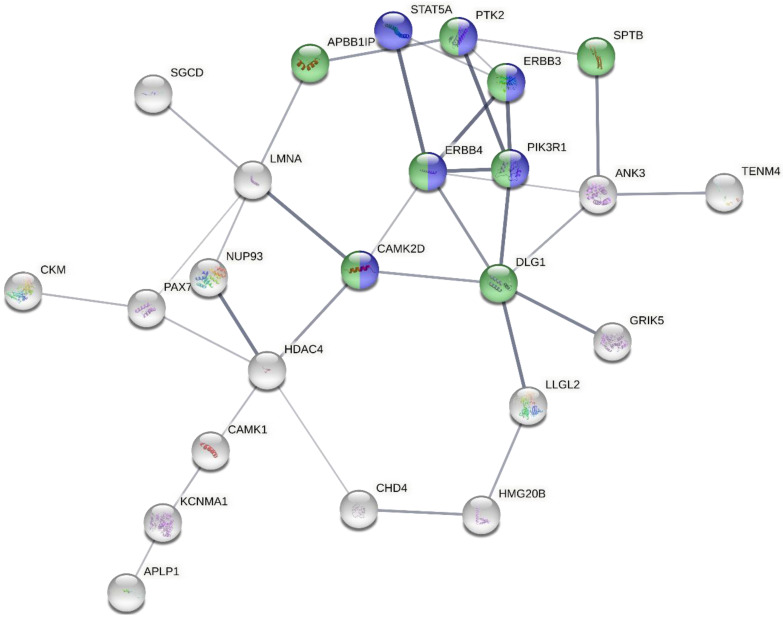
Protein interaction network of genes with upstream and 5′UTR variants generated by STRING. Green color in the nodes represents ErbB pathway involvement and blue represents the MAPK pathway. The density of the connecting lines between the protein nodes in the figure represents the interaction score, highlighting the importance of the MAPK and ErbB pathway-related proteins.

**Figure 4 cells-12-00096-f004:**
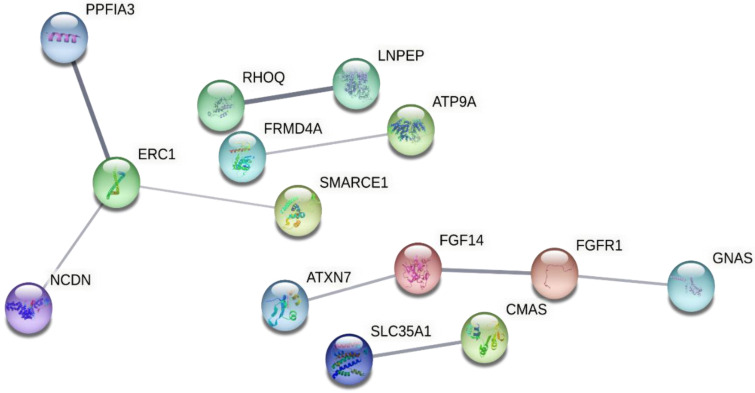
Protein interaction network of genes with 3′ UTR variants generated by STRING. No pathway enrichment was found here. Due to the lack of pathway detection, the color of the nodes was randomly selected by the software and serves only for differentiation purposes.

**Figure 5 cells-12-00096-f005:**
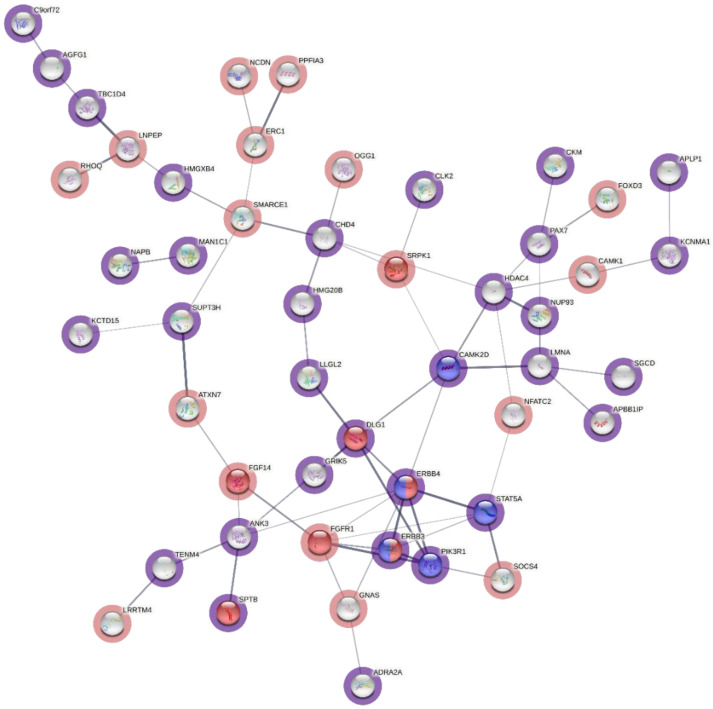
Protein interaction network of a combination of genes with upstream, 5′ and 3′ UTR variants generated by STRING. Proteins surrounded by violet circles represent genes with 5′ UTR variants and those surrounded by red circles represent genes with 3′ UTR variants; proteins with red filling belong to the MAPK pathway and those with blue filling belong to the ErbB pathway; some proteins, due to their involvement in both pathways, are filled with both red and blue. The density of the connecting lines between the protein nodes in the figure represents the interaction score, highlighting the importance of the MAPK and ErbB pathway-related proteins.

**Table 1 cells-12-00096-t001:** Upstream and 5′ UTR top and MAPK pathway variants prioritized on the basis of FCPPv2 and non-coding variant analysis tools within CADD v1.6, ENCODE TF data and Uniprot/Swiss-Prot functional information.

Family	Gene	Gene Name	Chrom_Pos_Ref_Alt	CADD	Conservation Score/3	CpG Island (yes/no)	Segway	cHmm	Histone Marks >20	No. of TFs	Conserved TFBSs	Encode TFs in GM12878/GM12878 ENCSR447YYN	Overall Function
								**>20%**					
Family_1	SP5 †	transcription factor Sp5	2_171571426_G_A	16.65	2	Yes	GS	TssA/TssAFlnk: Tx/TxFlnk/TxWk	EncH3K27Ac/K4Me1/K4Me3	13		NR2F1, HDAC6	DNA-binding transcription factor, bone morphogenesis, metal ion binding
Family_1	*FNDC3B* †	fibronectin type III domain containing 3B	3_171757553_C_A	15.8	2	Yes	TSS	TssA/TssAFlnk	EncH3K27Ac/K4Me3	40		BCLAF1, Yy1, Pax5, ETS1, TAF1, Tcf12, Egr1, POU2F2, ELF1, RUNX3	Adipogenesis
Family_1	*CAMK2D* *	calcium/calmodulin-dependent protein kinase II delta	4_114682943_TCCTCCTCCGGCG_T	19.58	3	No	TF2	ReprPC/RepPCWk/Quies	EncH3K27Ac/K4Me3	2		CTCF, BCL11A, EBF1, IRF4, BCLAF1, Pax5, Yy1, ELF1, TAF1, Egr1	Regulation of Ca^2+^ homeostasis
Family_1	FOXJ2 †	forkhead box J2	12_8185317_GGAGCC_G	21.9	2	Yes	TSS	TssA/TssAFlnk: TssBiv/EnhBiv	EncH3K27Ac/K4Me3	29		Egr1, SP1	Transcriptional activator
Family_1	SPTB *	spectrin, beta, erythrocytic	14_65346721_C_A	20.5	2	Yes	TSS	ReprPC/RepPCWk/Quies		NA	E47, Tal-1, ITF-2, Tal-1beta, GATA-1, AP-2alphaA, AP-2gamma	Egr1, HDAC6	Cytoskeleton network
Family_2	NRBF2 †	nuclear receptor binding factor 2	10_64893005_T_C	17.12	2	Yes	TSS	ReprPC/RepPCWk/Quies	EncH3K4Me3	1		ATF3, POU2F2, TAF1, ZBTB33, SP1, BCLAF1, Egr1, Tcf12, ELF1, Yy1	Autophagy, transcription regulation
Family_4	HMGXB4 †	HMG box domain containing 4	22_35653479_C_A	20.3	2	Yes	TSS	TssA/TssAFlnk	EncH3K27Ac/K4Me3	15	IRF-1	ELF1, ETS1, SP1, POU2F2, TAF1, BCLAF1, Yy1, Egr1, Tr4, Srf	Wnt signaling
Family_6	ERBB4 *	v-erb-a erythroblastic leukemia viral oncogene homolog 4 (avian)	2_213404066_C_T	17.09	2	No	D	TssA/TssAFlnk		7	p300	HDAC6	Tyrosine kinase, apoptosis, development
Family_6	AGFG1 †	ArfGAP with FG repeats 1	2_228337132_G_A	16.16	2	Yes	GS	TssA/TssAFlnk	EncH3K27Ac/K4Me3	8		Yy1, ELF1, BCLAF1, Pax5, Egr1, ETS1, BHLHE40, IKZF1, ZNF217, BACH1	Differentiation, mRNA transport
Family_6	ING2 †	inhibitor of growth family member 2	4_184425877_C_A	15.71	3	Yes	TSS	TssA/TssAFlnk	EncH3K4Me3	12		Yy1, BCLAF1, ELF1, Egr1, Tcf12, Pax5, SP1, POU2F2, Srf, MEF2A	Chromatin organization, histone deacetylation
Family_6	PIK3R1 *	phosphoinositide-3-kinase, regulatory subunit 1 (alpha)	5_67511017_G_C	16.81	1	Yes	TSS	Enh: ReprPC/RepPCWk/Quies	EncH3K27Ac/K4Me3	3		BCLAF1, ELF1, CTCF, MEF2A, Yy1, TAF1, Egr1, EBF1, Pax5, POU2F2	Protein transport, stress response
Family_6	MDFIC †	MyoD family inhibitor domain containing	7_114562322_C_G	21.1	2	Yes	TF0	TssBiv/Biv/EnhBiv:TssA/TssAFlnk	EncH3K27Ac/K4Me3	7		POU2F2, Egr1, BCLAF1, ETS1, Yy1, MEF2A, TAF1, ELF1, RB1, IKZF1	Transcription regulation, Wnt signaling
Family_6	TBC1D4 †	TBC1 domain family, member 4	13_76056522_G_A	18.11	2	Yes	GS	TssA/TssAFlnk:ReprPC/RepPCWk/Quies	EncH3K27Ac/K4Me3	1	NF-1	PU1, ELF1, POU2F2, Egr1, ETS1, Yy1, BCLAF1, CTCF, IRF4, Rad21	GTPase activator
Family_7	ERBB3 *†	v-erb-b2 erythroblastic leukemia viral oncogene homolog 3 (avian)	12_56473408_C_T	18.89	3	Yes	GS	TssA/TssAFlnk	EncH3K27Ac/K4Me1/K4Me3	69		CTCF, IKZF1, TRIM22, RB1, TCF3	Kinase, signal transduction regulation
Family_7	PSMC6 *†	proteasome (prosome, macropain) 26S subunit, ATPase, 6	14_53173885_C_G	18.51	3	Yes	TSS	TssA/TssAFlnk	EncH3K27Ac/K4Me3	13		Yy1, TAF1, POU2F2, ELF1, Srf, Gabp, SP1, SIN3A, RB1, PKNOX1, ZNF207, TBP, ELK1	Ubiquitination, immune system, Wnt signaling
Family_9	CAMK1 †	calcium/calmodulin-dependent protein kinase I	3_9811535_G_A	21.1	2	Yes	TSS	TssA/TssAFlnk:TssBiv/EnhBiv	EncH3K4Me3	18	Pax-5, MIF-1, AP-2gamma, USF1	RB1	Cell cycle, differentiation
Family_9	PLEKHG1 †	pleckstrin homology domain containing, family G (with RhoGef domain) member 1	6_150921086_G_A	15.37	3	Yes	TF0	TssA/TssAFlnk:TssBiv/EnhBiv	EncH3K4Me3	11		IKZF1, NR2F1, ZNF217 ELF1, BACH1, Tcf12 PU1, HDAC6, SP1	G nucleotide exchange factor
Family_10	PTK2/FAK1 *	protein tyrosine kinase 2/Focal Adhesion Kinase 1	8_142012766_C_T	15.65	1	No	GS	TssA/TssAFlnk	EncH3K27Ac/K4Me1	39			Cell cycle, migration, adhesion
Family_11	DLG1 *†	discs, large homolog 1 (Drosophila)	3_197024641_C_T	20.2	2	Yes	TSS	Enh	EncH3K27Ac/K4Me3	1		Yy1, Egr1, ELF1, POU2F2, TAF1, Tcf12, Pax5, HDAC6, ZNF24, BHLHE40	Host–virus interaction, cadherin binding
Family_11	APBB1IP *	amyloid beta (A4) precursor protein-binding, family B, member 1 interacting protein	10_26727608_C_G	15.3	2	Yes	TF0	ReprPC/RepPCWk/Quies	EncH3K27Ac/K4Me3	NA		Yy1, BCLAF1, Pax5, ELF1, PU1, Rad21, RUNX3, IKZF1 MEF2B, BACH1	Cell adhesion, immune system

* Genes identified through top variants in the FCVPPv2 analysis; † genes identifies by MAPK pathway. Chrom_Pos_Ref_Alt, chromosome,_position,_reference allele,_alternative allele; CADD: Combined Annotation Dependent Depletion; GS: gene start, TSS: transcription start site; TF: transcription factor; CHmm: Chrome HMM chromatin states; TssA: active transcription start site; TssFlnk: flanking transcription start site; TssBiv: bivalent/poised transcription start site; Tx: strong transcription; TxWk: weak transcription; TxFlnk: flanking transcription; Enh: enhancer; EnhBiv: bivalent enhancer; RepPC: repressed polycomb; RepPCWk: weak repressed polycomb; Quies: quiescent; Histone marks: Encode histone marks in 14 cell lines (only those in over 20% of cell lines are shown here); conservation scores: cumulative score of three different conservation scores (GerpN > 2, verPhyloP ≥ 3, verPhCons > 0.3), CpG island, Segway, chromatin states in 127 cell lines (those states are shown that are present in 20% or more cell lines), and the number of TFs/TFBSs is extracted from the annotation data on CADD website; Encode TFs in GM12878/GM12878 ENCSR447YYN is taken from SNP nexus. For variants that were predicted to affect the binding of more than 10 TFs, only 10 are shown in the table and a detailed list is given in Appendix A. The overall function of the genes is from Uniprot/SwissProt.

**Table 2 cells-12-00096-t002:** The 3′ UTR top and MAPK pathway variants prioritized based on FCPPv2 and non-coding variant analysis tools within CADD v1.6, and Uniprot/Swiss-Prot functional information.

Family	Gene	Gene Name	Chrom_Pos_Ref_Alt	CADD	Conservation Score	miRNA Binding yes/no	Mir SVR Score	Segway	cHmm > 20	Overall Function
						**(bold if context++>90)**				
Family_1	LONRF1 *	LON peptidase N-terminal domain and ring finger 1	8_12580093_G_C	19.75	3	**Yes**	−1.26	GE0	cHmm:Tx/TxWk	Protein polyubiquitination, metal ion binding
Family_2	SLC35A1 *	solute carrier family 35 (CMP-sialic acid transporter), member A1	6_88222026_A_G	16.88	2	**Yes**	−1.23	GE0	cHmm:Tx/TxWk	Transmembrane transport, carbohydrate metabolism
Family_6	MARCHF8 *	membrane-associated ring finger (C3HC4) 8, E3 ubiquitin-protein ligase	10_45952965_T_C	16.16	3	Yes	−0.78	GE0	cHmm:Tx/TxWk	Immune response, antigen processing MHC class II
Family_10	B4GALT5 *	UDP-Gal:betaGlcNAc beta 1,4- galactosyltransferase, polypeptide 5	20_48250790_A_G	16.44	3	Yes	−0.41	GE1	cHmm:Tx/TxWk	Galactosyltransferase, lipid metabolism, regulation of embryonic development
Family_12	FAM76B *	family with sequence similarity 76, member B	11_95504039_CA_C	15.82	3	**Yes**	−0.26	GE0	cHmm:Tx/TxWk	Unknown function
Family_13	SGSM2 *	small G protein signaling modulator 2	17_2284327_C_T	21.5	2	**Yes**	−1.22	GE1	cHmm:Tx/TxWk	GTPase activation, intracellular transport
Family_2	FGFR1 †	fibroblast growth factor receptor 1	8_38270114_C_T	17.46	1	Yes	−0.11	R5	cHmm:Tx/TxWk/ReprPC/PCWk/Quies	Cell migration, differentiation, proliferation, MAPK pathway

* Genes identified through top variants in the FCVPPv2 analysis; † genes identified by the MAPK pathway. Chrom_Pos_Ref_Alt: chromosome_position_reference allele_alternative allele; CADD: Combined Annotation Dependent Depletion; GE: gene end, R5: Repressed, CHmm: Chrome HMM chromatin states; Tx: strong transcription; TxWk: weak transcription; RepPC: repressed polycomb; RepPCWk: weak repressed polycomb, Quies: quiescent; conservation scores (cumulative score of three different conservation scores, GerpN > 2, verPhyloP ≥ 3, verPhCons > 0.3) is used here and details are given in Appendix A), Segway, chromatin states in 127 cell lines (those states are shown that are present in 20% or more cell lines) and mirSVR scores (−0.1 is considered significant) are extracted from annotation data on CADD website, miRNA binding and context ++ score (90 or above is considered high percentile score with high chance of miRNA affecting the mRNA) information is taken from Target Scan’s human miRNA atlas; overall function of the genes is from Uniprot/Swissprot.

**Table 3 cells-12-00096-t003:** Reactome pathway enrichment analysis for combined 5′ and 3′ UTR genes.

Reactome Pathway	Ratio of Proteins in Pathway	Number of Proteins in Pathway	Proteins from Gene Set	*p*-Value	FDR	Hit Genes
RAF/MAP kinase cascade	0.0253	276	10	1.19 × 10^−5^	3.88 × 10^−3^	*PIK3R1,PTK2,DLG1,FGFR1,SPTB,APBB1IP,CAMK2D,ERBB3,ERBB4,PSMC6*
MAPK1/MAPK3 signaling	0.0258	282	10	1.43 × 10^−5^	3.88 × 10^−3^	*PIK3R1,PTK2,DLG1,FGFR1,SPTB,APBB1IP,CAMK2D,ERBB3,ERBB4,PSMC6*
MAPK family signaling cascades	0.0298	326	10	4.88 × 10^−5^	8.78 × 10^−3^	*PIK3R1,PTK2,DLG1,FGFR1,SPTB,APBB1IP,CAMK2D,ERBB3,ERBB4,PSMC6*
Asparagine N-linked glycosylation	0.0262	286	9	9.90 × 10^−5^	0.01	*CMAS,ANK3,CTSA,NGLY1,SPTB,B4GALT5,NAPB,MAN1C1,SLC35A1*
PI3K events in ERBB2 signaling	0.0015	16	3	1.66 × 10^−4^	0.02	*PIK3R1,ERBB3,ERBB4*
Negative regulation of NMDA receptor-mediated neuronal transmission	0.0019	21	3	3.68 × 10^−4^	0.03	*DLG1,CAMK2D,CAMK1*
Long-term potentiation	0.0021	23	3	4.79 × 10^−4^	0.04	*DLG1,CAMK2D,ERBB4*
Signaling by ERBB4	0.0053	58	4	5.81 × 10^−4^	0.04	*PIK3R1,STAT5A,ERBB3,ERBB4*
Post NMDA receptor activation events	0.0057	62	4	7.43 × 10^−4^	0.04	*DLG1,CAMK2D,ERBB4,CAMK1*

FDR, false discovery rate.

## Data Availability

The data presented in this study are available on request from the corresponding authors. The data are not publicly available due to privacy reasons.

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
