# Peer review of "Investigation of Rare Non-Coding Variants in Familial Multiple Myeloma"

_cells, 2022, doi:10.3390/cells12010096_

Round 1

Reviewer 1 Report

see attached report

Author Response

Dear reviewer,

please find attached the response to the comments.

Reviewer 2 Report

This  highly interesting paper aims to investigate the molecular mechanisms and interactions at peptide level in the familial occurrence of myeloma which is a rare finding. The data and interpretation presented is a deep analysis on 14 families revealing  new non-coding gene variants conferring the predisposition to MM . Authors are reporting these variants to be found in pathways and genes previously implicated in MM risk, and thus reaffirm the involvement of ErbB and MAPK signaling pathways in MM pathogenicity.

Author Response

(The authors gave the same response as above.)

Reviewer 3 Report

In this study, the authors performed whole-genome sequencing of DNA from peripheral blood of 31 patients with multiple myeloma and 16 unaffected individuals in 14 myeloma families and identified 150 upstream 5’-UTR and 3’-UTR variants including 20 top scoring variants with 9 pathway variants. Overall, the analyses were carefully done and yielded convincing results. 

Author Response

Dear reviewer please find attached the response to the comments.

Round 2

Reviewer 1 Report

Authors have addressed my concerns in the revised version.